# DOES AURORA ENCODE ATMOSPHERIC STRUCTURE? LATENT REGIME ANALYSIS AND ATTRIBUTION

**Emma Kasteleyn**[*] **& Ana Lucic**
University of Amsterdam

## ABSTRACT

ML foundation models are able to emulate atmospheric dynamics accurately and efficiently but operate as opaque "black boxes". We investigate the internal representations of the Aurora model using spatially pooled PCA and layer-wise relevance propagation (LRP). We find evidence that Aurora's latent space is primarily organized by seasonal cycles, whereas extreme storm events do not form a linearly separable cluster. LRP indicates that the model attends to features consistent with the 3D vertical structure of the Great Storm of 1987. Perturbation tests show masking relevant regions degrades forecasts $3.31\times$ more than random masking. These findings suggest that Aurora learns meteorological coherence and vertical structure without explicit instruction. Code is available on GitHub.

## 1 INTRODUCTION

The field of weather forecasting is currently undergoing a paradigm shift comparable to the introduction of numerical methods in the 20th century. While numerical weather prediction (NWP) relies on explicitly solving systems of partial differential equations of fluid dynamics, recent deep learning (DL) models such as FourCastNet (Pathak et al., 2022), Pangu-Weather (Bi et al., 2022), Graphcast (Lam et al., 2023), and Aurora (Bodnar et al., 2024) learn atmospheric dynamics directly from reanalysis data. These models emulate dynamics at a fraction of the computational cost of operational systems like the ECMWF IFS (ECMWF, 2024), often with superior accuracy. However unlike NWP, where state updates are governed by physics-based models, DL models operate as "black boxes", encoding atmospheric states into high-dimensional, non-interpretable latent representations. This opacity creates an important barrier to operational trust: without understanding how a model arrives at a forecast, meteorologists cannot tell whether a model produces physically consistent predictions or relies on spurious correlations.

While explainable AI (XAI) has been successfully applied to meteorological CNNs (Ebert-Uphoff & Hilburn, 2020; Toms et al., 2020), the internal dynamics of billion-parameter foundation models remains a significant challenge (Yang et al., 2024). Richards & Balan (2025) utilized linear probes to analyze Aurora's encoder, finding that encoder embeddings capture concepts like land-sea contrast. Similarly, MacMillan & Ouellette (2025) applied sparse autoencoders (SAEs (Bricken et al., 2023)) to GraphCast. However, these approaches analyze encoder representations or graph-based message passing. Appplying propagation-based attribution to the deep self-attention backbone of a 3D transformer weather model remains unexplored. Standard perturbation methods (e.g., SHAP (Lundberg & Lee, 2017)) scale poorly to such high-dimensional spatiotemporal grids, necessitating efficient, propagation-based alternatives like Layer-wise Relevance Propagation (LRP (Bach et al., 2015)) to trace information flow.

In this work, we investigate the internal representations of Aurora, a foundation model based on a 3D Swin Transformer V2 (Liu et al., 2022) U-Net architecture. Unlike standard vision transformers (ViT (Dosovitskiy et al., 2021)), Aurora processes multi-variable atmospheric states through hierarchical shifted-window attention, learning general-purpose embeddings from heterogeneous datasets (e.g., ERA5 (Hersbach et al., 2023), HRES (ECWMF, 2024), CMIP6 (Scoccimarro et al., 2018)). Specifically, we structure our analysis around two core research questions: **RQ1:** Is the model's latent space organized by distinct meteorological regimes (e.g., seasonal cycles vs. dynamic storm

---

[*]Correspondence to: `emma.kasteleyn@student.uva.nl`

events)? and **RQ2:** Can the model's local attributions correctly isolate a particular storm system and reflect its 3D vertical structure?

To answer **RQ1**, we use spatially pooled PCA on the model's latent bottleneck, analyzing whether Aurora organizes atmospheric states into meaningful regimes. To address **RQ2**, we conduct a local case study on the Great Storm of 1987. By adapting LRP to the Swin backbone, we map the spatial and vertical drivers of the storm and validate these attributions via a single-shot perturbation test. Our results suggest that Aurora moves beyond simple pattern matching: it disentangles seasonal modes (**RQ1**) and encodes vertical atmospheric structure (**RQ2**) despite being a data-driven model.

## 1.1 DATA AND MODEL SETUP

We analyze the `AuroraSmallPretrained` foundation model, available via Hugging Face. The architecture is a 3D Swin Transformer V2 U-Net pre-trained on diverse atmospheric data (see Appendix A for architectural details). Inference is performed on $0.25°$ resolution ERA5 reanalysis data, normalized using the fixed statistics provided in the official model configuration. The model ingests atmospheric fields $\mathbf{X}_{\mathrm{in}} \in \mathbb{R}^{B \times T \times C_{\mathrm{phys}} \times H \times W}$ (Batch, Time, Channel [Physical Variable], Height, Width) and projects them into a compressed latent space.

## 2 LATENT REGIME ANALYSIS (RQ1)

To answer **RQ1**, we analyze how Aurora organizes atmospheric data in its latent space. We focus on the United States area (Lat $24°$–$50°$N, Lon $235°$–$295°$E; see Appendix B) and filter the 2005 hurricane season for extreme events. We targeted the top 30 days per regime, but strict thresholding ($v_{\max} \geq 24.0$ m/s for storms, $\leq 15.0$ m/s for calm) limited the dataset to $N_{\mathrm{storm}} = 17$ and $N_{\mathrm{calm}} = 30$. Additionally, we segment the 2005 dataset into meteorological seasons ($N \approx 91$ per season) to analyze the seasonal organization of the latent space.

Let $\mathbf{Z} \in \mathbb{R}^{B \times T \times H' \times W' \times D}$ be the latent feature map at the bottleneck of the Swin backbone, where $B$ is the batch size, $T$ is the number of timesteps, $H', W'$ are the spatially downsampled grid dimensions, and $D$ is the embedding dimension. Unlike the input space, $D$ jointly encodes the entire vertical column and all physical variables for a given location, making $D$ a good latent feature axis. We introduce a spatially pooled PCA strategy for latent structure analysis. For a defined target region mask $M_{PCA}$ (applied to dimensions $H', W'$), we compute a statistical descriptor vector $s_t \in \mathbb{R}^{3D}$ for each sample $b$ and timestep $t$ by concatenating the spatial statistics across the masked region:

$$s_t = \left[ \mathrm{mean}(\mathbf{Z}_t^{(M_{\mathrm{PCA}})}), \mathrm{std}(\mathbf{Z}_t^{(M_{\mathrm{PCA}})}), \mathrm{rms}(\mathbf{Z}_t^{(M_{\mathrm{PCA}})}) \right],$$

We apply singular value decomposition (SVD, (Eckart & Young, 1936)) to the standardized descriptors to identify dominant modes of variation. To explicitly test for regime separability beyond the principal components, we further compute a contrastive projection onto the difference vector $w = \mu_{\mathrm{storm}} - \mu_{\mathrm{calm}}$. We verify the robustness of these patterns via bootstrap resampling of the dataset ($N = 1000$) to quantify the stability of the resulting eigenvectors.

**Results.** We find that representations are primarily organized by the seasonal cycle rather than specific weather events. As shown in Fig. 1, the first principal component (PC1) captures 24.1% of the total variance, functioning as a seasonal axis that linearly separates winter and summer regimes. Bootstrap validation ($N = 1000$) demonstrates the stability of this latent geometry, yielding a cosine similarity of 0.998 for PC1 (Appendix Tab. 1). Fall and spring exhibit overlap, even in higher order components (Appendix Fig. 4). Similarly, the fall-spring contrastive projection exhibits overlap (Appendix Fig. 5), confirming that the model identifies structural similarities between these transitional seasons while maintaining a detectable separation in their mean representations.

Conversely, extreme weather events do not constitute a dominant linear mode. In Fig 1, storm samples are distributed without distinct clustering. The contrastive projection confirms this, revealing that storm and calm regimes form distinct but overlapping distributions (Appendix Fig. 5). This interpretation is reinforced by stability analysis: while the seasonal PC3 is stable ($\mu = 0.977$), the storm-regime PC3 exhibits significant instability ($\mu = 0.650, \sigma = 0.269$). The high variance likely results from the combination of limited sample size and the intrinsic spatial heterogeneity of individual cyclone tracks, which prevents the emergence of a stable higher-order direction.

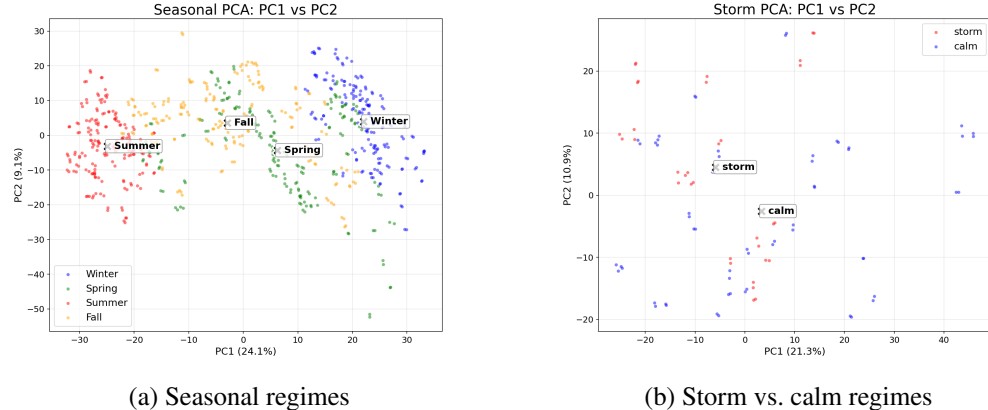

(a) Seasonal regimes          (b) Storm vs. calm regimes

Figure 1: **PCA Analysis.** (a) Seasons show distinct clustering, unlike (b) storms.

## 3   LOCAL ATTRIBUTION AND VERTICAL STRUCTURE (Q2)

To answer **Q2**, we apply LRP to the Great Storm of 1987, a classic extratropical cyclone. We address the specific architectural challenge of the Swin V2 backbone – shifted window self-attention – by wrapping the attention mechanism to preserve the computational graph during cyclic spatial shifts (torch.roll) (see Appendix B for implementation details). We utilize the `Zennit` framework (Anders et al., 2023) with the LRP-$\epsilon$ rule ($\epsilon = 0.25$) for linear/convolutional layers.

To isolate the drivers of specific forecasts for the target region, we define a binary spatial mask $M_{\text{LRP}} \in \{0,1\}^{H' \times W'}$ corresponding to the target region in the latent grid, in this case, Europe (Lat $30°–72°$N, Lon $−12°–45°$E). We initialize the relevance propagation by computing the Hadamard product of the latent activations and the mask: $\mathbf{R} = \mathbf{Z} \odot M_{\text{LRP}}$. This approach is analogous to analyzing the effective receptive field (Luo et al., 2016). By setting relevance to zero everywhere outside the target region, we ensure that the backward pass only tracks the information flow contributing to the model's representation of that specific area. This yields input-space heatmaps that identify the input features responsible for the model's representation of the target region.

To quantify the faithfulness of the generated explanations, we perform a single-shot perturbation analysis, adapted from the evaluation strategy of Samek et al. (2015). While our qualitative results focus on visualizing the European target region, we define our quantitative perturbation search space as the broader North Atlantic sector (Lat $20°–80°$N, Lon $300°–45°$E). Since regional latent targeting propagates globally, using this wider window acts as a selectivity test: it verifies that the method successfully isolates the relevant storm system rather than selecting irrelevant signals from the upstream Atlantic sector. We mask the top $k = 1\%$ of pixels with the highest relevance scores within this sector and measure impact via the forecast distortion ratio:

$$\mathcal{D} = \frac{\text{MSE}(f(\mathbf{X}_{\text{LRP}}), f(\mathbf{X}))}{\text{MSE}(f(\mathbf{X}_{\text{rand}}), f(\mathbf{X}))}$$

where $f(\cdot)$ is Aurora's prediction of mean sea level (MSL) pressure over the target region. $\mathcal{D} > 1.0$ shows that the LRP-identified top $1\%$ acts as a stronger driver of representations than random noise.

**Results.** For the storm (Fig. 2), relevance for temperature, wind, and MSL pressure is tightly localized to the cyclone's head and frontal boundaries, constrasting with the baseline, where relevance is different per variable and geographically locked to static features like topography (e.g., Scandinavia, Appendix Fig. 6). Importantly, vertical analysis (Fig. 3) shows that activations for surface wind contribute to latent features encoding the upper troposphere ($\approx 150$ hPa), suggesting that Aurora has learned to link surface anomalies with their necessary vertical drivers despite being data-driven.

Masking the top $1\%$ of LRP-identified pixels degrades the latent reconstruction significantly more than random masking ($MSE_{LRP} = 5.30 \times 10^4$ vs. $MSE_{rand} = 1.60 \times 10^4$). The resulting impact ratio of $3.31\times$ confirms that LRP successfully isolates the features affecting the model's inference.

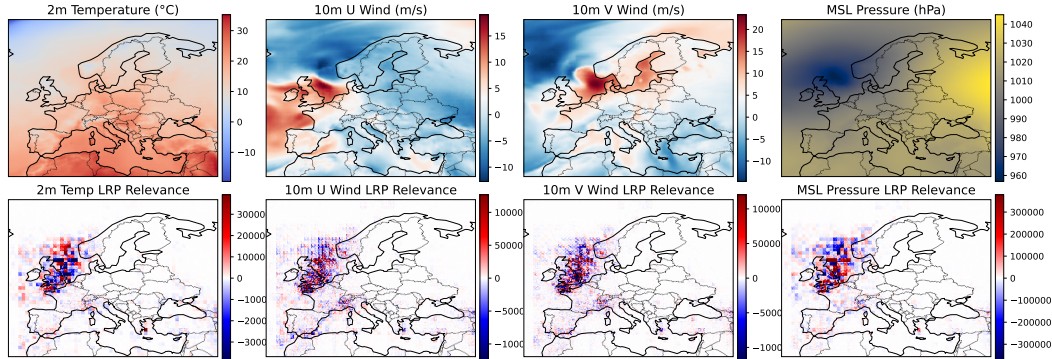

Figure 2: **Surface variable relevance.** 1987 Great Storm: Model focuses on dynamic frontal features.

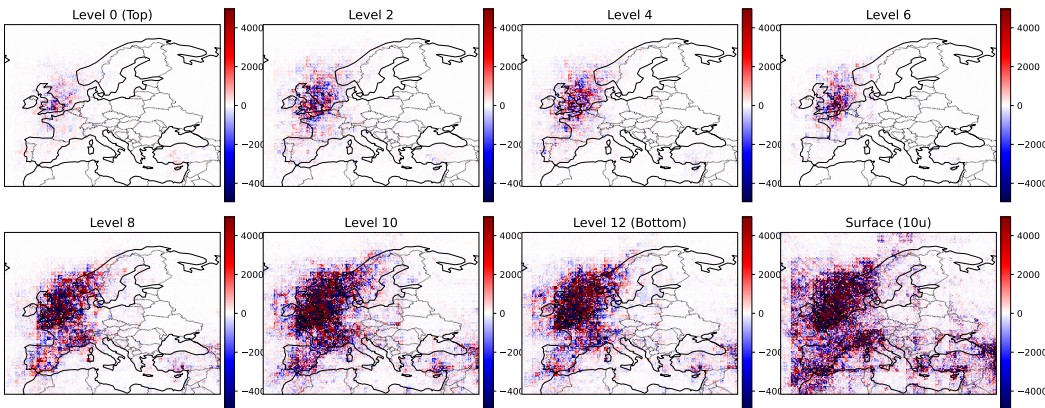

Figure 3: **Atmospheric variable relevance.** 1987 Great Storm: Model captures frontal boundaries on multiple levels.

## 4 CONCLUSION

This study investigated the internal representations of the Aurora foundation model through two complementary lenses. Regarding global organization (**RQ1**), results suggest that Aurora captures the annual seasonal cycle, while storms are not encoded in the first levels of PCA. LRP analysis suggests that Aurora is capable of capturing the 3D vertical structure of the Great Storm of 1987 (**RQ2**). The model links surface wind anomalies to the upper troposphere ($\approx 150$ hPa) despite being a purely data-driven model. Our perturbation analysis indicates that the model does not rely on static geographical shortcuts, but prioritizes physically relevant features.

While promising, this initial audit has limitations. (1) We operate on the small version of Aurora due to computational constraints, future work should focus on the large model. (2) Our definition of storm relies solely on scalar wind magnitude ($v_{max}$). Future work should incorporate vector metrics (e.g., vorticity) to distinguish between unstructured wind bursts and coherent, rotating systems. (3) Our latent analysis was performed on a relatively small sample size, and it is possible that the observed overlap in seasons reflects the limitations of linear PCA in resolving the cyclic topology of the seasonal signal. While distinct physical states typically are separable regions of the high-dimensional manifold (Modell et al., 2025), linear projections inevitably merge opposing phases of a cycle (e.g., spring vs. fall). (4) Our attribution analysis is limited to one particular storm using LRP, which serves as a promising starting point, but future work should evaluate many instances of diverse meteorological phenomena and explore more efficient attention-based attribution methods, such as LeGrad (Bousselham et al., 2025).

ACKNOWLEDGEMENTS

This work was performed using the computational resources of the Snellius supercomputer, provided by the University of Amsterdam. We thank the ELLIS Unit Amsterdam for their support and for facilitating this research as part of the ELLIS MSc Honours Programme.

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

## A  THE AURORA FOUNDATION MODEL ARCHITECTURE

Aurora (Bodnar et al., 2024) is a large-scale foundation model designed to learn general-purpose atmospheric representations from heterogeneous Earth-system data. Its architecture combines a Perceiver IO (Jaegle et al., 2021) interface, which ingests multi-variable inputs at arbitrary resolutions, with a 3D Swin Transformer V2 (Liu et al., 2022) U-Net backbone that evolves atmospheric states over time.

### A.1  ENCODER AND LATENT REPRESENTATION

The encoder addresses the high dimensionality and variable resolution of weather data by projecting diverse inputs into a unified latent tensor. This process involves three distinct transformations relevant to our feature analysis:

- **Patching and Embedding:** The gridded field for each meteorological variable is partitioned into patches of size $P \times P$. Each patch is linearly projected to an embedding vector.

- **Vertical Compression (Latent Perceiver):** Atmospheric variables often span a variable number of pressure levels. Aurora employs a Perceiver module to compress the vertical column into a fixed number of latent levels (typically $L = 8$). This yields a consistent 3D representation suitable for downstream processing.

- **Fourier Positional Encodings:** Instead of learned positional embeddings, Aurora uses Fourier features to encode spatial location, area, and time. This provides explicit geometric grounding and enables the model to operate at arbitrary resolutions without retraining.

### A.2  3D SWIN TRANSFORMER BACKBONE

The latent state is propagated through a 3D Swin Transformer V2 backbone. Unlike standard global attention, Swin architectures use local self-attention within non-overlapping windows, enabling linear scaling with input size. Aurora adapts this for geophysical data:

- **Attention Mechanism.** It retains Res-Post-Norm for stability but employs standard dot-product attention rather than scaled cosine attention.

- **Positional Bias.** The model discards the relative position biases typical of Swin models, relying entirely on the absolute Fourier encodings injected by the encoder. This suggests that the attention heads attend to absolute physical locations rather than relative pixel distances.

- **Memory optimization.** To handle the high memory footprint of 3D atmospheric states, the model implementation leverages activation checkpointing (Chen et al., 2016) and ZeRO-based optimizations (Rajbhandari et al., 2020).

### A.3  PRE-TRAINING OBJECTIVE

The model was pre-trained on a massive corpus of heterogeneous data, including ERA5 reanalysis (Hersbach et al., 2023) and HRES operational forecasts (ECMWF, 2024), for approximately 150k steps using a Mean Absolute Error (MAE) objective. This pre-training forces the model to learn a compressed, physically consistent representation of atmospheric dynamics, which we probe in this study via the fixed weights of the `AuroraSmallPretrained` checkpoint.

## B  EXTENDED METHODOLOGY AND IMPLEMENTATION DETAILS

### B.1  DERIVED VARIABLES AND DATA PREPROCESSING

**Wind Magnitude Calculation.** To construct the regime-dependent datasets (storm vs. calm), we derive the scalar wind magnitude from the raw vector components provided by the ERA5 reanalysis. Let $\mathbf{X}_{\text{in}} \in \mathbb{R}^{C \times H \times W}$ represent the input tensor for a single time step. We denote the indices for the 10m Zonal ($U$) and Meridional ($V$) wind components as $c_u$ and $c_v$ respectively. The wind magnitude

map $\boldsymbol{M}_v \in \mathbb{R}^{H \times W}$ is computed element-wise:

$$M_v^{(i,j)} = \sqrt{(\mathbf{X}_{\text{in}}^{(c_u,i,j)})^2 + (\mathbf{X}_{\text{in}}^{(c_v,i,j)})^2}$$

The event classification metric $v_{\text{max}}$ is defined as the maximum value within this bounding box. For storm events we select $v_{\text{max}} \geq 24$ m/s, as this corresponds with Beaufort scale 10, classifying the event as a storm (Royal Meteorological Society, n.d.).

**Numerical Stabilization.** To ensure numerical stability during LRP (Binder et al., 2016), we register a backward hook on the input tensor to intercept the backward flow. This hook zeros out NaN values that occasionally arise from the $\epsilon$-stabilized denominator in "silent" regions of the atmosphere, preventing numerical singularities from corrupting the final heatmap.

## B.2 LRP Specifics

We implement LRP to satisfy the conservation property $\sum_i R_i = \sum_j R_j$, where relevance is redistributed from the output neuron back to the input pixel space.

**Composite Rules.** We utilize the `Zennit` framework (Anders et al., 2023) to define a composite rule set that handles the heterogeneous layers of the Swin Transformer V2 U-Net. The relevance propagation rules $R_j = \sum_k \frac{z_{jk}}{\sum_j z_{jk}} R_k$ are parametrized as follows:

- **Convolutional and Linear Layers:** We apply the LRP-$\epsilon$ rule with a stabilizer term $\epsilon = 0.25$ to dampen noise and prevent numerical instability when activations approach zero:

$$R_j = \sum_k \frac{a_j w_{jk}}{\epsilon + \sum_h a_h w_{hk}} R_k$$

- **Normalization Layers (LayerNorm):** Treated as identity operators for relevance flow to preserve the magnitude of attribution passing through the backbone's frequent normalization stages.

- **Skip Connections:** Relevance is distributed equally across the split branches, summing at the merge point during the backward pass.

**Attention Mechanism and Cyclic Shifts.** The Swin Transformer V2 uses shifted window self-attention, relying on cyclic shifts (torch.roll) to facilitate cross-window connections. While standard automatic differentiation computes gradients through these shifts correctly, LRP requires an explicit inverse mapping to ensure relevance conservation. We wrap the attention mechanism to ensure that the backward hook for the roll operation maps relevance $R_{\text{shifted}}$ back to $R_{\text{original}}$ indices exactly, preventing misalignment between the attribution map and the physical spatial grid.

## B.3 Statistical Robustness

**Bootstrap Validation.** To estimate the confidence intervals of the PCA eigenvectors, we employ non-parametric bootstrapping. Given the dataset $\mathcal{D} = \{\boldsymbol{s}_1, \ldots, \boldsymbol{s}_N\}$ of size $N$, we generate $B = 1000$ resampled datasets $\mathcal{D}_b^*$ by sampling $N$ items from $\mathcal{D}$ with replacement. We re-compute the SVD for each $\mathcal{D}_b^*$ to obtain perturbed eigenvectors $\boldsymbol{v}_b^*$. Stability is quantified by the cosine similarity between the original principal component $\boldsymbol{v}$ and the bootstrap mean $\bar{\boldsymbol{v}}^*$:

$$\text{Stability} = \frac{\boldsymbol{v} \cdot \bar{\boldsymbol{v}}^*}{\|\boldsymbol{v}\|\|\bar{\boldsymbol{v}}^*\|}$$

**Perturbation Thresholding.** For the stability metric $\mathcal{D}$, we enforce a strict sparsity constraint ($k = 1\%$) on the perturbation mask. We deliberately adopt a strict 1% threshold to assess the attribution of critical features; higher perturbation percentages (e.g., $> 5\%$) were found to introduce widespread stochastic noise that disrupts the transformer's patch embedding structure independently of feature importance.

## C ADDITIONAL RESULTS

### C.1 BOOTSTRAP RESULTS

Table 1: Bootstrap Stability ($N = 1000$). Mean cosine similarity ($\pm$SD) of PCA components.

| Component | Mean Cosine Sim. | Std. Dev. |
|---|---|---|
| **Storm regime** | | |
| PC1 | 0.975 | $\pm$ 0.017 |
| PC2 | 0.909 | $\pm$ 0.066 |
| PC3 | 0.650 | $\pm$ 0.269 |
| **Seasonal regime** | | |
| PC1 | 0.998 | $\pm$ 0.001 |
| PC2 | 0.986 | $\pm$ 0.012 |
| PC3 | 0.977 | $\pm$ 0.015 |

### C.2 HIGHER-ORDER PCA

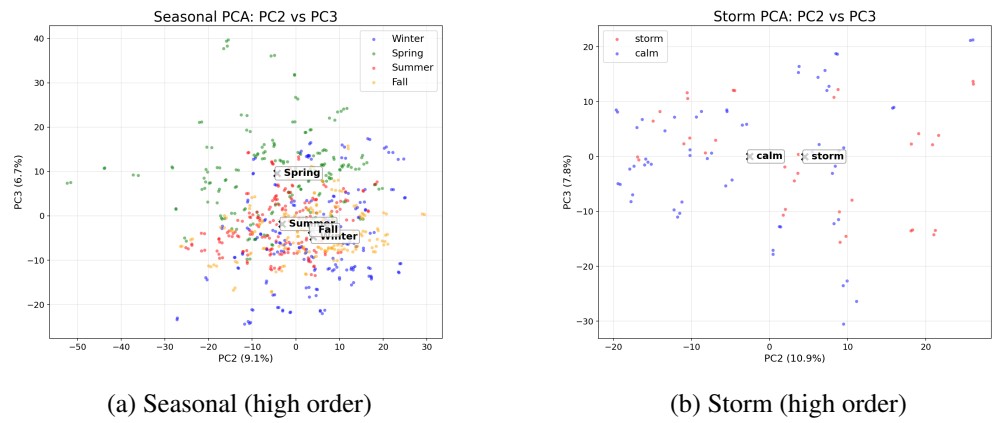

(a) Seasonal (high order)  (b) Storm (high order)

Figure 4: **Higher-Order Latent Components (PC2 vs. PC3).** (a) Seasonal clusters become less distinct. (b) Storm/calm regimes show no separability.

### C.3 CONTRASTIVE PROJECTION

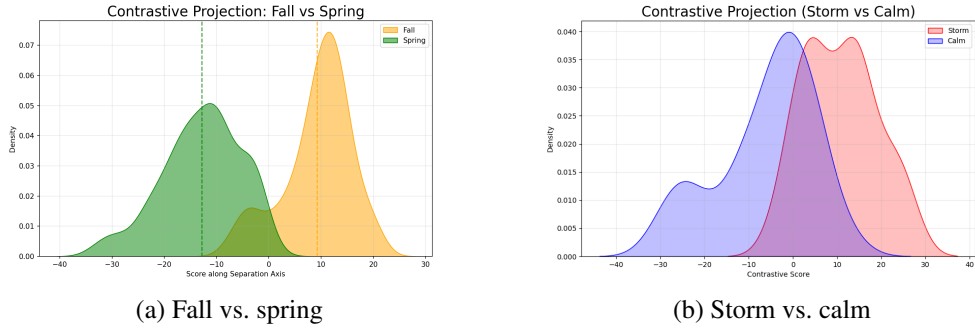

(a) Fall vs. spring  (b) Storm vs. calm

Figure 5: **Contrastive projections.** (a) Fall–spring projection showing shared support region. (b) Storm–calm projection showing partial separation.

## C.4 BASELINE SURFACE LRP

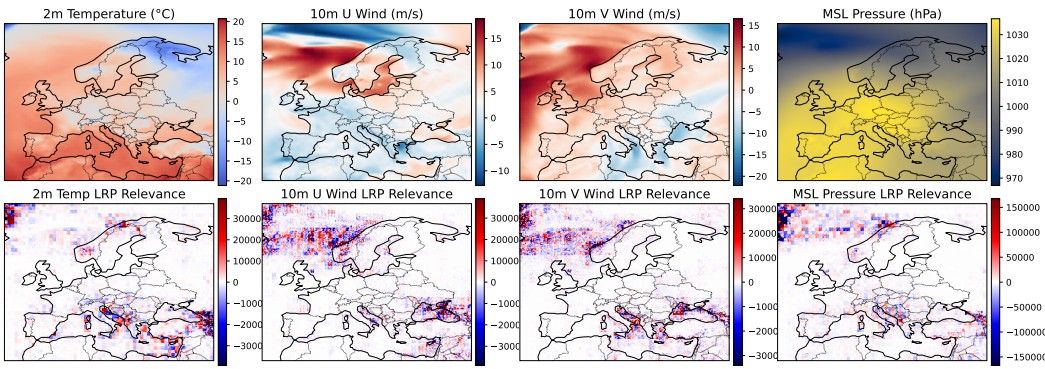

Figure 6: **LRP Relevance Maps.** 2020 Baseline (1 Jan): Model focuses shifts to static geography.

## C.5 BASELINE LEVELS LRP

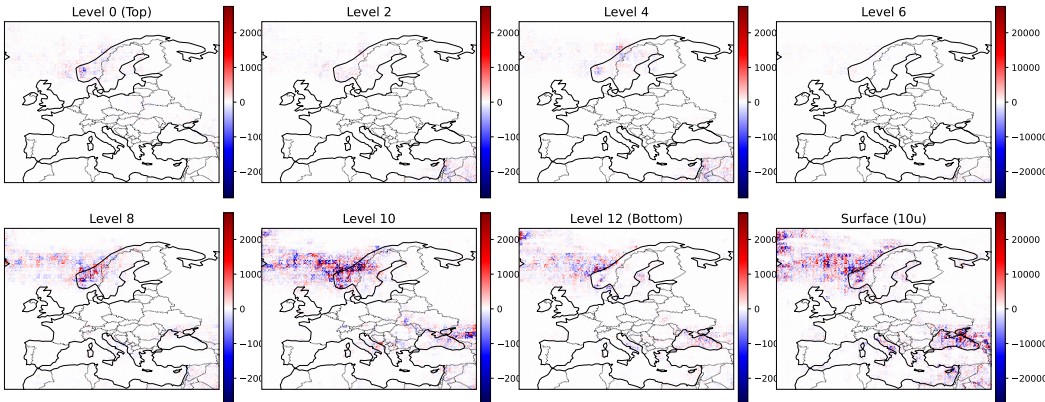

Figure 7: **Surface U-Wind Relevance.** 2020 Baseline (1 Jan): Model mainly captures wind relevance on lower levels.

