# OpenReview forum: "Does Aurora Encode Atmospheric Structure? Latent Regime Analysis and Attribution"
_ICLR.cc/2026/Workshop/FM4Science — ICLR 2026 Workshop FM4Science Poster_

### Official Review · Reviewer_U8jw · 2026-02-21
**An initial interpretability audit of the Aurora weather foundation model showing that its latent space is largely organized by seasonal structure**

**Rating:** 4
**Confidence:** 3

**Review:**

## 1. Summary
This paper is an interpretability focused audit of the Aurora weather foundation model. It studies (RQ1) whether Aurora’s latent space encodes meaningful meteorological regimes by applying a spatially pooled PCA to the bottleneck latent features, and (RQ2) whether local attributions identify physically relevant storm structure by adapting Layer-wise Relevance Propagation (LRP) to Aurora’s Swin Transformer backbone and evaluating relevance via a perturbation test. The core findings are that latent organization appears dominated by seasonal cycle structure, while storm events do not form a clean separable cluster; and that LRP highlights cyclone-relevant frontal features and exhibits higher impact under targeted masking than random masking.

## 2. Strengths and Weaknesses
### a) Soundness
**Strengths.** The paper poses clear questions and applies two complementary methodologies (global latent analysis + local attribution). The perturbation test is a reasonable faithfulness check, and the authors also note numerical and architectural challenges (cyclic shifts in Swin attention) and address them.

**Weaknesses.** The latent PCA approach is somewhat coarse: spatially pooled mean/std/rms descriptors may wash out localized dynamical regimes, and linear PCA is not well matched to cyclic/curved manifolds (which the authors themselves acknowledge). The storm-vs-calm dataset is also small (Nstorm=17) under their thresholding, which weakens statistical power for regime conclusions.

### b) Presentation
**Strengths.** The writing is concise and the study is easy to follow. The paper does a good job explaining how the relevance initialization is localized to a target region and why the perturbation sector is broader.

**Weaknesses.** Several choices would benefit from stronger motivation: why this particular pooling statistic, why PCA rather than (say) nonlinear embeddings or representation similarity measures, and why only one storm case study for LRP beyond the baseline day comparison. These gaps do not invalidate the work but limit interpretability breadth.
### c) Significance
**Strengths.** Interpretability of atmospheric foundation models is a meaningful problem for trust and scientific use. A practical LRP adaptation for Swin-based weather models plus a faithfulness check is potentially useful to the community.

**Weaknesses.** The scope is modest: one model checkpoint, one main attribution case study (1987 Great Storm), and relatively coarse latent analysis. This reads more like an initial audit than a comprehensive regime/structure study.

### d) Originality
**Strengths.** Adapting propagation-based attribution (LRP) to a Swin Transformer weather backbone and combining it with a regime-structure analysis is a useful and somewhat novel methodological combination in this niche.

**Weaknesses.** Many elements are established in XAI; novelty hinges on the Aurora-specific adaptation and framing. To strengthen originality, the paper would benefit from broader regime taxonomy (beyond storm/calm and seasons), or from comparing multiple XAI methods to isolate what LRP uniquely provides.

## 3. Key Questions for Authors
1.	How sensitive are your conclusions to the pooling design (mean/std/rms) and to the choice of region mask? Would a token-level or spatially localized analysis reveal storm separability that pooled statistics erase?

2.	Given Nstorm=17, can you report uncertainty bands or power analysis for storm/calm separability? Would relaxing thresholds (or using cyclone tracking / vorticity-based labels) materially change the conclusion?
3.	The Great Storm of 1987 is one case. Have author(s) tested whether the dynamic frontal feature relevance pattern holds across multiple storms and seasons? If not, how should readers interpret generality?

4.	The perturbation metric uses top 1% masking within a sector. How robust is the 3.31x ratio to different masking fractions, and does it persist if you perturb in physically meaningful ways (e.g., coherent patches, not pixels)?

## 4. Limitations
The paper explicitly discusses limitations such as storm definition, sample size, linear PCA on cyclic signals, computational overhead of LRP and suggests future directions.

Refer to the weaknesses and questions as my justification of the rating.

---

### Official Review · Reviewer_4vqc · 2026-02-23
**Review of the paper "Does Aurora Encode Atmospheric Structure?"**

**Rating:** 5
**Confidence:** 3

**Review:**

**Summary** - The paper investigates what atmospheric foundation models actually learn internally, using the Aurora weather model as a case study.  Rather than improving forecasting performance, the paper asks - *RQ1:* Is Aurora’s latent space organized by meaningful meteorological regimes? *RQ2:* Do internal attributions correspond to physically realistic atmospheric structure?

**Strengths** -
1. Interpretability of foundation weather models is a high-impact open problem.
2. Perturbation validation is interesting. RQ1 and RQ2 are precise and easy to follow.
3. Applying LRP to shifted-window 3D Swin attention is interesting.

**Weaknesses** -
1. Most conclusions rely on one hurricane season, limited storm samples, one detailed storm case study. This is insufficient to support broad claims about foundation model representations. The authors also acknowledge small sample size limitation explicitly. I believe, extensive evaluation in the form of - multi-year validation, multiple storm types, cross-basin analysis, etc. would be helpful in validating the conclusions.
2. The experiments utilize linear PCA to analyze the latent geometry. But atmospheric dynamics are generally cyclic, nonlinear. So, the key conclusion, that "storms are not encoded" may be an artifact of linear projection. This weakens RQ1 substantially.
3. Missing baselines. The paper show LRP works, but that is not compared to any baseline approach. Some common comparisons could be, attention rollout, gradient × input, or integrated gradients. Currently, we cannot conclude whether LRP is uniquely meaningful, or any attribution would look similar.

---

### Decision · Program_Chairs · 2026-03-03

Accept (Poster)